# SWITCHING-ALIGNED-WORDS DATA AUGMENTATION FOR NEURAL MACHINE TRANSLATION

## ABSTRACT

In neural machine translation (NMT), data augmentation methods such as back-translation make it possible to use extra monolingual data to help improve translation performance, while it needs extra training data and the in-domain monolingual data is not always available. In this paper, we present a novel data augmentation method for neural machine translation by using only the original training data without extra data. More accurately, we randomly replace words or mixup with their aligned alternatives in another language when training neural machine translation models. Since aligned word pairs appear in the same position of each other during training, it is helpful to form bilingual embeddings which are proved useful to provide a performance boost (Liu et al., 2019). Experiments on both small and large scale datasets show that our method significantly outperforms the baseline models.

## 1 INTRODUCTION

Deep neural networks show great performances when trained on massive amounts of data. Data augmentation is a simple but effective technique to generate additional training samples when deep learning models are thirsty for data. In the area of Computer Vision, it is a standard practice to use image data augmentation methods because trivial transformations for images like random rotation, resizing, mirroring and cropping (Krizhevsky et al., 2012; Cubuk et al., 2018) doesn't change its semantics. This presence of of semantically invariant transformation makes it easy to use image data augmentation in Computer Vision research.

Unlike image domain, data augmentation on text for Natural Language Processing (NLP) tasks is usually non-trivial as there is often a prerequisite to do some transformations without changing the meaning of the sentence. In this paper we will focus on data augmentation techniques in neural machine translation (NMT) which is special and more difficult than other NLP tasks since we should maintain semantic consistency within language pairs which is from quite possibly different domains.

Data augmentation techniques in NMT can be divided into two categories dependent on whether additional monolingual corpus is uesd. If in-domain monolingual training data for NMT is available, one successful data augmentation method is back-translation (Sennrich et al., 2016), whereby an NMT model is trained in the reverse translation direction (target-to-source) and then used to translate target-side monolingual data back to source language. The resulting synthetic parallel corpus can added to existing training data to learn a source-to-target model. Other more refined ideas of back-translation include dual learning (He et al., 2016) or Iterative Back-translation (Hoang et al., 2018).

Sometimes when in-domain monolingual data is limited, existing methods including randomly swapping two words, dropping word, replacing word with another one (Lample et al., 2018) and so on are applied to perform transfromations to original training data without changing its semantics to the greatest extent. However, due to text characteristics, these random transformations often result in significant change in semantics. Gao et al. (2019) propose to replace the embedding of word by a weighted combination of mutiple semantically similar words. Also, Xiao et al. (2019) use a lattice structure to integrate multiple segmentations of a single sentence to perfrom an immediate data augmentation.

In this work, we propose Switching-Aligned-Words (SAW) data augmentation, a simple yet effective data augmentation approach for NMT training. It belongs to the second class of data augmentation

methods where in-domain monolingual data is limited. Different from the previous methods that conduct semantically invariant transformations within each language, we propose to use another language (target language) to help make semantically invariant transformations for current language (source language) by switching aligned words randomly. We use an unsupervised word aligner *fast-align*[1] (Dyer et al., 2013) to pair source and target words that have similar meaning.

To verify the effectiveness of our method, we conduct experiments on WMT14 English-to-German and IWSLT14 German-to-Englisth datasets. The experimental results show that our method can obtain remarkable BLEU score improvement over the strong baselines.

## 2 RELATED WORK

We describes the related work about data augmentation for NMT with or without using additional monolingual data in this section.

### 2.1 WITH MONOLINGUAL DATA

The most successful data augmentation techiques to leverage monolingual data for NMT training is back-translation. It requires training a target-to-source system in order to generate additional synthetic parallel data from the monolingual target data. This data complements human bitext to train the desired source-to-target system. There has been a growing body of literature that analyzes and extends back-translation. Edunov et al. (2018) demontrate that it is more effective to generate source sentences via sampling rather than beam search. Hoang et al. (2018) present iterative back-translation, a method for generating increasingly better synthetic parallel data from monolingual data to train NMT model. Fadaee & Monz (2018) show that words with high predicted loss during training benefit most. Wang et al. (2019) propose to quantify the confidence of NMT model predictions based on model uncertainty to better cope with noise in synthetic bilingual corpora produced by back-translation. Dual learning (He et al., 2016) extends the back-translation approach to train NMT systems in both translation directions. When jointly training the source-to-target and target-to-source NMT models, the two models can provide back translated data for each other direction and perform multi-rounds back-translation.

Different from back-translation, Currey et al. (2017) show that low resource language pairs can also be improved with synthetic data where the source is simply a copy of the monolingual target data. Wu et al. (2019) propose to use noised training to better leverage both back-translation and self-training data.

### 2.2 WITHOUT MONOLINGUAL DATA

Lample et al. (2018) randomly swap the words within a fixed small window size or drop some words in a sentence for learning an autoencoder to help train the unsupervised NMT model. Fadaee et al. (2017) propose to replace a common word by low-frequency word in the target sentence, and change its corresponding word in the source word to improve translation quality of rare words. In Xie et al. (2017), they replace the word with a placeholder token or a word sampled from the frequency distribution of vocabulary, showing that data noising is an effective regularizer for NMT. Kobayashi (2018) propose an approach to ues the prior knowledge from a bi-directional language model to replace a word token in the sentence. Gao et al. (2019) try to replace the ids of word by a soft ids and they train Transformer language models in original training data to get soft words. Wang et al. (2018) introduce a data augmentation method for NMT called SwitchOut to randomly replace words in both source and target sentences with other words.

## 3 OUR APPROACH

We first describe the background and our proposed switching-aligned-words data augumentation approach. The framework can be seen as an adversarial training process like Generative Adversarial Networks (GAN) (Goodfellow et al., 2014; Salimans et al., 2016), see Figure 1 for an overview. For

---

[1]https://github.com/clab/fast_align

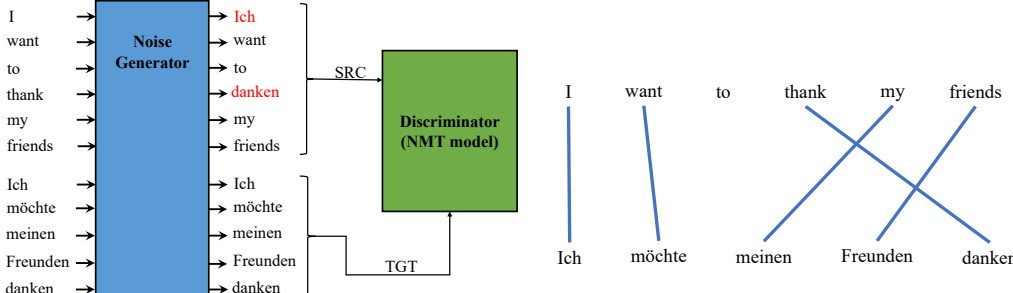

Figure 1: An overview of Switching-Aligned-Words data augumentation approach. The noise generator can be any model that produces noise over parallel sentences, and the NMT model is trained as a discriminator.

Figure 2: The illustration for alignment model. English sentence is *"I want to thank my friends."*, and corresponding German sentence is *"Ich möchte meinen Freunden danken"*.

image generation, in which a discriminator and a generator compete with each other: the generator aims to generate images similar to the natural ones, and the discriminator aims to detect the generated ones from the natural ones. For data augmentaion methods in NMT, the noise generator can be any model that produces noise over parallel sentences, in our method it is an alignment model which is shown in Figure 2. Finally, the NMT model is trained as a discriminator to distinguish generated sentences from the original ones and the process of detection noise offers NMT model an ability to learn bilingual alignment information.

## 3.1 BACKGROUND

Given a source and target sentence pair $(\boldsymbol{x}, \boldsymbol{y})$, where $\boldsymbol{x} = (x_1, x_2, \cdots, x_{|\boldsymbol{x}|})$ is a source-language sentence and $\boldsymbol{y} = (y_1, y_2, \cdots, y_{|\boldsymbol{y}|})$ is a target-language sentence. A neural machine translation system models the conditional probability:

$$P(\boldsymbol{y}|\boldsymbol{x}) = \prod_{j=1}^{|\boldsymbol{y}|} P(y_j|\boldsymbol{y}_{<j}, \boldsymbol{x}) \tag{1}$$

based on an encoder-decoder framework with an attention mechanism (Sutskever et al., 2014; Bahdanau et al., 2014). Encoder and decoder can be specialized using different neural architectures including GRU (Bahdanau et al., 2014), LSTM (Wu et al., 2016), CNN (Gehring et al., 2017) and Transformer (Vaswani et al., 2017), among which the self-attention based Transformer is the state-of-the-art architecture for NMT.

The decoder predicts a corrresponding translation $\boldsymbol{y} = (y_1, \cdots, y_{|\boldsymbol{y}|})$ step by step based on the last decoding state and source context. The translation probability can be formulated as follows:

$$P(y_j|\boldsymbol{y}_{<j}, \boldsymbol{x}) = q(y_{j-1}, s_j, c_j) \tag{2}$$

where $s_j$ and $c_j$ denote the decoding state and the source context at the j-th time step respectively. Here, $q(\cdot)$ is the softmax layer. Sepcifically,

$$s_j = g(y_{j-1}, s_{j-1}, c_j) \tag{3}$$

where $g(\cdot)$ is the corresponding neural architecture unit. The context vector $c_j$ is calculated as a weighted sum of the source annotations $h_i$ on the basis of attention mechanism:

$$c_j = \sum_{i=1}^{|\boldsymbol{x}|} \alpha_{ji} h_i \tag{4}$$

The alignment model $\alpha_{ji}$ measures the similarity between $s_j$ and $h_i$. The whole model is jointly trained to seek the optimal parameters that can be used to correctly encode the source sentences and decode them to corresponding target sentences.

### 3.2 ALIGNMENT

NMT models learn the alignment between source words $x_i$ and target word $y_j$ mainly deponds on these two aspects: attention and word embeddings. Since attention weight $\alpha_{ji}$ measures the similarity between $s_j$ and $h_i$, it has been widely used to evaluate the word alignment between $y_j$ and $x_i$, so that the word alignment is explicitly modeled.

NMT models also try to learn word alignment information by updating word embeddings when training. In monolingual vector space, similar words tend to have commonalities in the same dimensions of their word vectors (Mikolov et al., 2013). These commonalities include: (1) a similar degree (value) of the same dimension and (2) a similar positive or negative correlation of the same dimension. In bilingual vector space, Liu et al. (2019) assume that the source and target words that have similar meanings should also have similar embedding vectors. Hence, they propose to perform a sharing techique between source and target word embedding space resulting significantly imporvement in alignment quality and translation performance.

Motivated by their findings, we propose to generate new training samples by replacing one word in the original sentences with its alinged word in corresponding target sentences. According to the characteristic of bilingual embeddings, aligned words tend to have similar meanings even in different language, so our replacing method will preserve the original meaning of the sentence to a great extend. Also, when training the model we put a aligned target word in the similar context of source sentence, it is helpful for source and target words with similar meanings to learn similar embedding representation.

### 3.3 SWITCHING ALIGNED WORDS BY REPLACEMENT

Inspired by the above intuition, we propose to augment NMT training data by replacing a randomly chosen word in a sentence by its aligned target word. Suppose we have an extra alignment model $A(\cdot|\cdot)$ such as intrinsic attention mechanism (Bahdanau et al., 2014) or unspervised word aligner (Dyer et al., 2013). Given a sentence pair $(\boldsymbol{x}, \boldsymbol{y})$, each source word $x_i$ is aligned with a target word $\hat{y}_i$ that has the highest alignment probability among the candidates, and is computed as follows:

$$\hat{y}_i = \underset{y \in a(x)}{\arg\max} \, log A(y|x_i) \tag{5}$$

where $a(\cdot)$ denotes the set of aligned candidates. So the conditional probability can be written as:

$$
\begin{aligned}
P(\boldsymbol{y}|\boldsymbol{x}) &= \prod_{j=1}^{|\boldsymbol{y}|} P(y_j|\boldsymbol{y}_{<j}, C(\boldsymbol{x})) \\
&= \prod_{j=1}^{|\boldsymbol{y}|} P(y_i|\boldsymbol{y}_{<j}, x_1, \dots, \hat{y}_k, \dots, x_{|\boldsymbol{x}|})
\end{aligned}
\tag{6}
$$

where $k$-th source word is replaced by corresponding target word. In experiments, we randomly choose a word in the training data with probability $\gamma_1$ and replace it by its aligned target word.

### 3.4 SWITCHING ALIGNED WORDS BY MIXUP

Mixup is a simple yet effective image augmentation techique introduced by Zhang et al. (2017). The idea is to combine two random images in a mini-batch in some proportion to generate synthetic examples for training. Bringing this idea to our work, we do not directly replace source word with corresponding aligned target word with probability $\gamma_1$, instead we mix up these two word embeddings to form a combined embedding which contain both source and target information:

$$
\begin{aligned}
\boldsymbol{E}(x_i) &= (1 - \gamma_2)\boldsymbol{E}(x_i) + \gamma_2 \boldsymbol{E}(C(x)) \\
&= (1 - \gamma_2)\boldsymbol{E}(x_i) + \gamma_2 \boldsymbol{E}(\hat{y}_i)
\end{aligned}
\tag{7}
$$

where $\boldsymbol{E}$ is the embedding lookup table, $\gamma_2$ is the mixup ratio which is a hyper-parameter.

The intuition behind mixup is that random linear interpolations between the embeddings of source word and corresponding target word let neural models regularize the representation of word embeddings. Mixing the aligned word pairs do not interrupt the representaion of word embeddings far from its original ones.

# 4 EXPERIMENT

In this paper, data augmentation will only process source data of the training data.

## 4.1 DATASETS

Two translation tasks, IWSLT14 German-to-English (De-En) and WMT14 English-to-German (En-De), are used for our evaluation.

**IWSLT14 German-English** IWSLT14 De-En dataset contains 153K training sentence pairs. We randomly select 7K data from the training set as validation set and use the combination of dev2010, dev2012, tst2010, tst2011 and tst2012 as test set with 7K sentences which are preprocessed firstly. BPE algorithm is used to process words into subwords, and number of subword tokens in the shared vocabulary is 10k.

**WMT14 English-German** We use the WMT14 En-De dataset with 4.5M sentence pairs for training. We randomly select 40K data from the training set as validation set and use newstest2014 as test set. Dataset is segmented by BPE and the number of subword tokens in the shared vocabulary is 32K. The sentences longer than 250 subword tokens are removed from the training dataset.

## 4.2 BASELINES

We compare our approach with following baselines:

- **Base**: The original training strategy without any data augmentation;
- **Swap**: Randomly swap words in nearby positions with a window size k (Lample et al., 2018);
- **Dropout**: Randomly drop word tokens (Lample et al., 2018);
- **Blank**: Ramdomly replace word tokens with a placeholder token (Xie et al., 2017);
- **Smooth**: Randomly replace word tokens with a sample from the unigram frequency distribution over the vocabulary (Xie et al., 2017);

All above introduced methods except **Swap** incorporate a hyper-parameter, the probability $\gamma$ of each word token to be replaced in training phase. We set $\gamma$ with different values in 0,0.05,0.1,0.15,0.2, and report the best result for each method. As for **Swap**, we use 3 as window size following (Lample et al., 2018);

## 4.3 MODEL

We use the *transformer_base* setting following Vaswani et al. (2017) for WMT14 En-De datasets, with a 6-layer encoder and 6-layer decoder. The dimensions of word embeddings, hidden states and the position-wise feed-forward networks are 512, 512, 2048 respectively. The dropout is 0.1 and attention head is 8. For IWSLT14 De-En datasets, we use the *transformer_small* setting which has a 6-layer encoder and 6-layer decoder, but the dimensions of word embeddings, hidden states and the position-wise feed-forward networks are 512, 512, 1024 respectively. The dropout is 0.3 and attention head is 4. Word embeddings between the source, target and output softmax embeddings are tied as it is a normal setting. We set $\gamma_1$ and $\gamma_2$ with different values in $\{0, 0.05, 0.1, 0.15, 0.2\}$, and report the best result for each method. For all experiments, hyperparameters are optimized on a development set and then tested using only a single hyperparameter. We use beam size 4 and length penalty 0.6 for inference, and use *multi-bleu*[2] to evaluate the quality of translation.

## 4.4 TRAINING

All our models are trained on one TITAN RTX GPU. The implementation of model is based on fairseq toolkit[3]. We choose Adam optimizer with $\beta_1 = 0.9$, $\beta_2 = 0.98$, $\epsilon = 10^{-9}$ and the learning

---

[2]https://github.com/moses-smt/mosesdecoder/blob/master/scripts/generic/multi-bleu.perl
[3]https://github.com/pytorch/fairseq

| Model | BLEU | |
|---|---|---|
| | **DE-EN** | **EN-DE** |
| Transformer (small) | 34.49 | - |
| Transformer (base) | - | 27.35 |
| +Swap | 34.40 | 27.12 |
| +Dropout | 34.83 | 27.43 |
| +Blank | 34.93 | 27.52 |
| +Smooth | 34.98 | 27.50 |
| **+Replacement** | 35.18 | 27.74 |
| **+Mixup** | 34.96 | 27.68 |

Table 1: BLEU scores on IWSLT14 De-En and WMT14 En-De. The baselines for De-En task and En-De task are the Transformer-small and the Transformer-base model respectively.

rate setting strategy, which are all the same as Vaswani et al. (2017), $lr = d^{-0.5} \cdot min(step^{-0.5}, step \cdot warmup_{step}^{-1.5})$ where $d$ is the dimension of embeddings, $step$ is the step number of training and $warmup_{step}$ is the step number of warmup. When the number of step is smaller than the step of warmup, the learning rate increases linearly and the decreases. Significantly, our replacing or mixing decision is made at runtime allowing different transformations for the same sentence pair.

## 4.5 RESULTS

The evalution results on IWSLT14 De-En and WMT14 En-De datasets are shown in Table 1. As we can see, the *Replacement* method can achieve 0.69 and 0.39 BLEU scores improvement over the Transformer small and the Transformer base baselines and the *Mixup* method improve the two baselines by 0.47 and 0.33 BLEU scores respectively.

Compared with other augmentation methods, we can see that (1) the *Replacement* method achieves the best results on all the datasets and (2) the *Mixup* method can achieve comparable or better results. Specially, we find that our method works better on relatively small scale datasets. As small scale datasets lack bilingual information compared to large scale datasets and are easy to fall into the overfitting problems, these results clearly demonstrate the effectiveness of our approach.

## 5 STUDY

### 5.1 IMPACT OF $\gamma_1$ AND $\gamma_2$

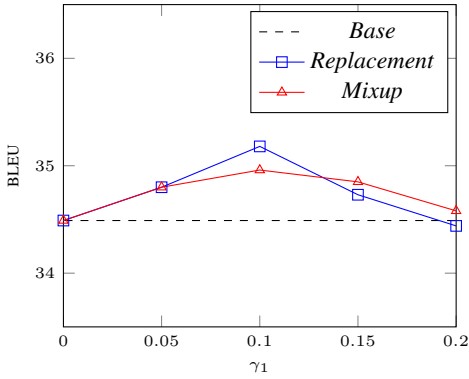
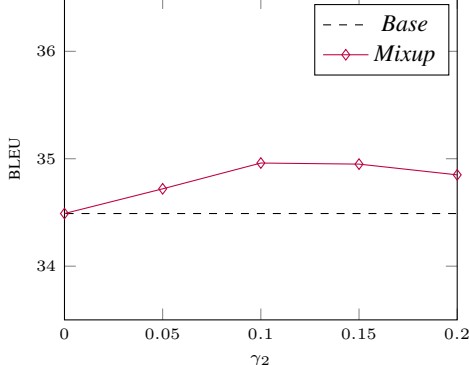

Figure 3: BLEU scores on IWSLT De-En dataset with different replacing probability $\gamma_1$. In *Mixup* experiment $\gamma_2$ is 0.1.

Figure 4: BLEU scores on IWSLT De-En dataset with different mixup probability $\gamma_2$ when $\gamma_1 = 0.1$.

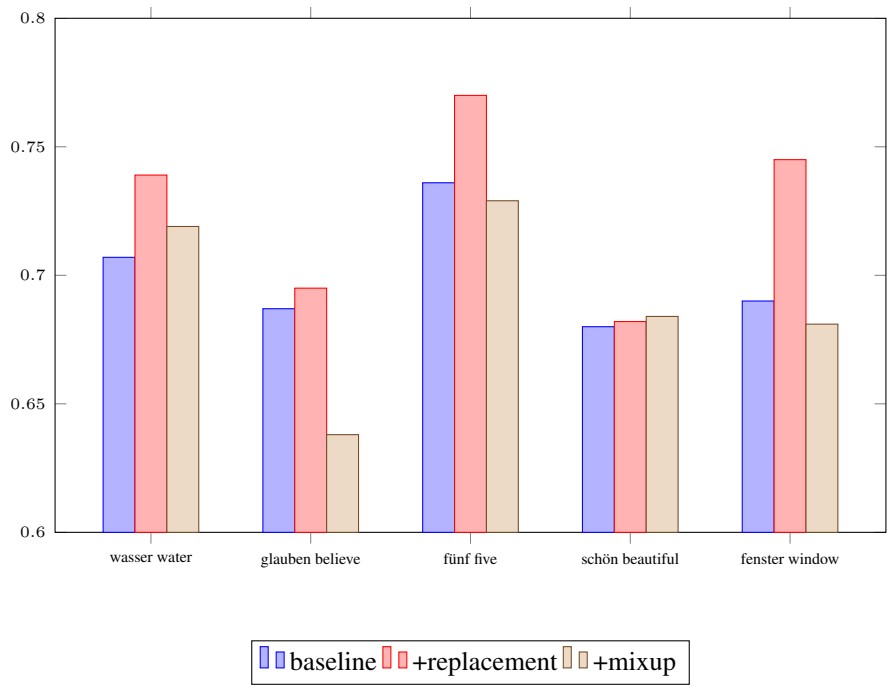

Figure 5: Cosine similarity between some bilingual embedding pairs in different method (the results have be normalized to 0 and 1).

We set different replacing probability value $\gamma_1$ and mixup probability value $\gamma_2$ to see the effect of our approach.

Figure 3 shows the BLEU scores on IWSLT14 De-En dataset of each method with different replacing probability, from which we can see that our method can obtain a consistent BLEU improvement within a large probability range and achieve the best performance when $\gamma_1 = 0.1$ in each method. However, the performance begins to drop when $\gamma_1 > 0.1$, we think the reason is that the semantic meanings of original sentence begin to be destroyed greatly. Also we find that *Mixup* is more stable than *Replacement*.

As we can see from Figure 4, the *Mixup* method can obtain a consistent BLEU improvement above baseline within a large probability range and the best BLEU socre is achieved in mixup probability $\gamma_2 = 0.1$ when $\gamma_1 = 0.1$.

## 5.2 Analysis of Bilingual Embeddings

Since we suppose that aligned word pairs appear in the same position of each other during training will be helpful to form bilingual embeddings which are proved useful to provide a preformance boost (Liu et al., 2019), we study whether our approach is truly useful for bilingual embeddings. We randomly sample some words and their corresponding aligned words to analyze the relation within them. Specifically, we compare the cosine similarity between the embeddings of aligned words to figure out the changes of bilingual embeddings. Formally, we have aligned word pairs $(\boldsymbol{x}_i, \boldsymbol{y}_j)$ and their embeddings $\boldsymbol{E}(\boldsymbol{x}_i) = (\boldsymbol{e}(\boldsymbol{x}_i)_1, \boldsymbol{e}(\boldsymbol{x}_i)_2, \cdots, \boldsymbol{e}(\boldsymbol{x}_i)_d), \boldsymbol{E}(\boldsymbol{y}_j) = (\boldsymbol{e}(\boldsymbol{y}_j)_1, \boldsymbol{e}(\boldsymbol{y}_j)_2, \cdots, \boldsymbol{e}(\boldsymbol{y}_j)_d)$, where $d$ is the embedding dimension. The cosine similarity can be defined as:

$$\cos\theta_{(\boldsymbol{E}(\boldsymbol{x}_i), \boldsymbol{E}(\boldsymbol{y}_j))} = \frac{\sum_{k=1}^{d} \boldsymbol{e}(\boldsymbol{x}_i)_k \cdot \boldsymbol{e}(\boldsymbol{y}_j)_k}{\sqrt{\sum_{k=1}^{d} \boldsymbol{e}(\boldsymbol{x}_i)_k^2} \cdot \sqrt{\sum_{k=1}^{d} \boldsymbol{e}(\boldsymbol{y}_j)_k^2}} \tag{8}$$

where $\theta_{(\boldsymbol{E}(\boldsymbol{x}_i), \boldsymbol{E}(\boldsymbol{y}_j))}$ is the angle between embedding pairs. We finally normalize the results to 0 and 1, and the larger the value, the more similar the two embeedings are.

From Figure 5 we can see that (1) The embedding vectors between aligned word pairs have a very strong positive correlation since the normalized cosine similarity values are all above 0.5. (2) The

*Replacement* method significantly imporves the positive correlation between aligned word pairs which proves our hypothesis that switching aligned words is helpful to from bilingual embeddings. (3) The *Mixup* method does not seem to improve the quality of bilingual embeddings. We suppose that the improvement of translation quality mainly come from the introduction of noise to word embeddings.

## 6 CONCLUSION

In this work, we have presented Switching-Aligned-Words (SAW) data augmentaion for NMT, which randomly replace words or mixup with their aligned alternatives in another language when training. It is simple yet effective and can be extremely useful when extra in-domain monolingual data is limited. Results on both small and large scale datasets have verified the effectiveness of our method.

In the future, besides focusing bilingual machine translation tasks, we are interested in extending our method to a multilingual scenario which needs more complex replacement and training strategies. In addition, we plan to study our approach in other cross-lingual NLP tasks.

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
