# OpenReview forum: "Switching-Aligned-Words Data Augmentation for Neural Machine Translation"
_ICLR.cc/2021/Conference — Reject_

### Official Review · AnonReviewer3 · 2020-10-28
**Interesting idea, but the experiments are not sufficient to provide a good picture of the effectiveness of the proposed approach.**

**Rating:** 4
**Confidence:** 5

**Review:**

Summary:

The paper proposes a GAN process for training neural machine translation models. The noise generator in this approach uses a switching-aligned-words technique where they randomly switch a word in the source sentence with its translation in the target sentence. They use fast-align to get alignments between source and target sentences. The experiments show that the noisy sentence pair generator performs best with the proposed switch and align approach in comparison with other (more random) methods.


##########################################################################

Pros:

Using GANs for training an NMT model is an interesting idea. I like the idea of using word translations as replacements for data augmentation. Since the embeddings of these words are close, these can be good candidates for the noise generator to fool the discriminator.

The paper covers the literature quite well.

##########################################################################

Cons:

The intuition of the paper is not clearly defined.

The experiments do not cover other augmentation methods. In this paper, the authors compare their approach to the following baselines: SWAP, DROPOUT, BLANK, SMOOTH. However, these are baseline approaches for the noise generator component. It would be valuable to know how this approach performs in comparison with other augmentation approaches such as [1] and [2].

All baseline methods (SWAP, DROPOUT, BLANK, SMOOTH) have an element of randomness in them and are not strong baselines. This provides little insight for understanding the impact of the bilingual switching.

The lack of comparison with the literature makes it difficult to evaluate the reported results.

It would be insightful to cover at least another language pair where the relation between the source and the target language is different. For instance, two languages that are not similar at all (structurally, morphologically, or semantically). It's interesting to see how this model performs when the assumption that word embeddings of the same word in two different languages are close to each other.

#########################################################################

Some typos:

(1) Typo on page 1: additional monolingual corpus is uesd -> additional monolingual corpus is used

(2) Typo in Figure 1: SRT -> SRC

(3) Typo on page 4:  words with similar meandings  ->  words with similar meanings

(4) Typo on page 7: we have aligned words pairs -> we have aligned word pairs

(5) Typo on page 7: switching aligned words is helpeful -> switching aligned words is helpful

#########################################################################

References:

[1] Rico Sennrich, Barry Haddow, and Alexandra Birch. Improving neural machine translation models with monolingual data. In ACL, 2016.

[2] Marzieh Fadaee, Arianna Bisazza, and Christof Monz. Data augmentation for low-resource neural machine translation. In ACL, 2017.

---

> ### Author Response · Authors · 2020-11-17
> **Response to Reviewer3**
>
> Thanks for your insightful comments.
>
> 1) The intuition behind the paper is that since the word and its corresponding word in the other language have similar embeddings, we can replace words with their aligned counterparts to add noise to source sentences but maintain semantic close. Our experiments prove it a good intuition for data augmentation.
>
> 2) Our settings are on any scale datasets and without using any monolingual data of source language, while [1] uses extra monolingual source data and [2] targets low-resource settings.
>
> 3) We are currently running replications for each experiment.
>
> 4) We will add more experiments on different language pairs.
>
> 5) Typos have been corrected.

---

### Official Review · AnonReviewer1 · 2020-10-29
**Unclear whether this idea represents a significant improvement over existing techniques.**

**Rating:** 4
**Confidence:** 4

**Review:**

This paper describes a method for data augmentation and/or regularization for machine translation that works by running a word aligner on the parallel data, and then with some probability \gamma, replacing a source token with its corresponding target token or vice versa. A proposed variant also mixes the embeddings of the two words. Small improvements are shown over simpler noising strategies such as replacing words with placeholder tokens or with random words from the vocabulary.

This is a neat idea, I like the motivation that the corresponding word in the other language is likely to maintain semantic coherence (though at the cost of linguistic fluency). However, the results are simply not strong enough to warrant a strong recommendation. Table 1 shows that the proposed method performs about the same as the stronger baselines (+0.2 or +0.24 BLEU). Furthermore, with both the method and the baselines having a substantial random component, I strongly urge the authors to carry out several random replications of each experiment, so we can get error bars around these results, and perhaps carry out a replication-aware significance test, such as [1]. As it is, no attempt to do significance testing is made.

I also think this work is missing some details: I assume that the swapping decision is made at runtime (and not at data construction time), allowing different swaps for the same sentence pair depending on the epoch, but it would be nice to be clear about this. I also assume that the authors have implemented shared source, target and output softmax embeddings, so as to strengthen the argument that equivalent words in different languages should have similar embeddings, but again, this should be made clear.

I also think the authors missed some relevant baselines -- these are perhaps equivalent to some of the other baselines mentioned, but they should be discussed [2] [3] [4].

I did not find the analogy to GANs or Figure 1 to be particularly useful for understanding this method.

Finally, there are some spelling errors that indicate that a spell-checker wasn’t run:

Garman → German

negrator → generator

[1] Jonathan Clark, Chris Dyer, Alon Lavie, and Noah Smith, "Better Hypothesis Testing for Statistical Machine Translation: Controlling for Optimizer Instability", Proceedings of the Association for Computational Lingustics, 2011. https://github.com/jhclark/multeval

[2] SwitchOut: an Efficient Data Augmentation Algorithm for Neural Machine Translation
Xinyi Wang, Hieu Pham, Zihang Dai, Graham Neubig, EMNLP 2018, https://arxiv.org/abs/1808.07512

[3] Provilkov I, Emelianenko D, Voita E. BPE-dropout: Simple and effective subword regularization. ACL 2020. https://arxiv.org/abs/1910.13267

[4] Robust Neural Machine Translation with Doubly Adversarial Inputs
Yong Cheng, Lu Jiang, Wolfgang Macherey. ACL 2019. https://arxiv.org/abs/1906.02443

---

> ### Author Response · Authors · 2020-11-17
> **Response to Reviewer1**
>
> Thanks for your insightful comments.
>
> 1) We are currently running replications for each experiment.
>
> 2) Our swapping decision is made at runtime allowing different swaps for the same sentence pair. We have tied word embeddings between the source, target and output softmax embeddings as it is a normal setting. We have added these details to the paper according to your advice.
>
> 3) Thank you for your advice, we have added discussions about [2].  [3] introduces a new subword regularization method called BPE-Dropout to overcome the problem that BPE splits words into unique subword sequences. [4] uses doubly adversarial inputs to improve the robustness of the NMT systems. These two works are orthogonal to our work but not very closely related to ours.
>
> 4) Since we switch words randomly in the training phase, the NMT training is like a GAN process. The noise generator produces noise where several words in the source sentence are randomly replaced by its translation in the target sentence, and the NMT model is trained as a discriminator. Because the word embeddings of these word pairs are close, it can be a good noise generator to fool the discriminator.
>
> 5) Spelling errors have been corrected.

---

### Official Review · AnonReviewer4 · 2020-10-29
**Interesting trick for training NMT systems**

**Rating:** 3
**Confidence:** 4

**Review:**

This paper shows that aligning parallel text with fastalign and then randomly replacing source words with their aligned target words, or interpolating their embeddings, improves machine translation.

This method is different from other data-augmentation methods that try to alter the source sentence without changing its meaning; here, the source sentence is altered into a mixture of the source and target. That’s interesting, but not very strongly motivated.

The paper doesn’t make clear whether the noise probability / coefficient is optimized on a development set or the test set. Based on Figures 3 and 4, it looks as though these hyperparameters may have been optimized on the test set, which is concerning. For both the baseline systems and your system, hyperparameters should be optimized on a development set and then tested using only a single hyperparameter setting on the test set. If this is what you did, please explicitly state this to reassure the reader.

Not much attempt is made to explain why this method helps; the only analysis is a measurement of cosine similarity between five German-English word pairs. Do you tie word embeddings between the source and target languages (Press and Wolf, 2017)?

- If so, one would expect that the transformer would already be able to place words with similar meanings close together, so the fact that your method improves this is interesting; do you know whether it helps more, e.g., for rare words, proper names, technical terms? Why is fastalign able to align some words better than the transformer? Would an even simpler method help, e.g., if (and only if) word f and word e both occur <= k times in the training data and they occur in exactly the same sentence pairs, then allow f to be switched to e?

- If not, I'd suggest doing so and rerunning the experiments to see if you still get an improvement.

Overall, this seems like a good trick for training NMT systems, but I would hope to see more insight either into why the proposed method works, or how NMT works or doesn’t work.

---

> ### Author Response · Authors · 2020-11-17
> **Response to Reviewer4**
>
> Thanks for your insightful comments.
>
> 1) For all experiments, hyperparameters are optimized on a development set and then tested using only a single hyperparameter. We are sorry that we did not clear it explicitly and we will add it to the revised version of the paper.
>
> 2) We have tied word embeddings between the source, target and output softmax embeddings as it is a normal setting. Even the Transformer would already be able to place words with similar meanings close together, our experiments show that our methods bring the embeddings of them closer compared to the Transformer baseline.
>
> 3) We will test the cosine similarity between word pairs in different frequencies.

---

### Official Review · AnonReviewer2 · 2020-11-01
**The paper proposes a data augmentation technique where source sentences are perturbed to include a few aligned words from the target language.**

**Rating:** 2
**Confidence:** 5

**Review:**

The paper proposes a data augmentation technique where source sentences are perturbed by replacing (or mixing) source words with their aligned counterparts from the target language (while the target sentences remain as is). Alignments can be either obtained from an unsupervised aligner like fast-align or from the attention distribution of an NMT model. Perturbations are aimed to be semantically invariant to preserve the meaning of the source sentence. In addition to simply replacing the source word with the aligned word, authors also try out inputting a weighted combination of both the source word and the target word and refer to this method as “mixing”. Empirical observations suggest that simply replacing the source word with the aligned target yields better results.

I believe the paper in the current form has a lot of scope for improvement. The experimental section needs to be strengthened and more thought needs to go into improving the proposed method.

I have the following suggestions for improving the paper:
More experiments: The experiments are based on a reasonably large translation dataset, while the paper "claims" to improve the performance for low-resource NMT in section 4.5

> Specially, we find that our method works better on a low resource settings

---

> ### Author Response · Authors · 2020-11-17
> **Response to Reviewer2**
>
> Thanks for your insightful comments.
>
> 1) We have modified our expression from "Specially, we find that our method works better on low resource settings" to "Specially, we find that our method works better on relatively small scale datasets".
>
> 2) We are running more experiments on different language pairs and smaller scale datasets.

---

### Decision · Program_Chairs · 2021-01-07
**Final Decision**

**Decision:**

Reject

**Comment:**

All reviewers agreed to reject.